# Effectiveness of Different Final Irrigation Procedures on *Enterococcus faecalis* Infected Root Canals: An In Vitro Evaluation

**DOI:** 10.3390/ma15196688

**Published:** 2022-09-27

**Authors:** Sanda Ileana Cîmpean, Ioana-Sofia Pop-Ciutrila, Sebastian-Roberto Matei, Ioana Alina Colosi, Carmen Costache, Gheorghe Zsolt Nicula, Iulia Clara Badea, Loredana Colceriu Burtea

**Affiliations:** 1Department of Conservative Dentistry and Endodontics, Iuliu Hatieganu University of Medicine and Pharmacy, 33 Motilor Street, 400001 Cluj-Napoca, Romania; 2Department of Molecular Sciences, Discipline of Microbiology, Iuliu Hatieganu University of Medicine and Pharmacy, 6 Louis Pasteur Street, 400349 Cluj-Napoca, Romania; 3Department of Molecular Sciences, Discipline of Cell and Molecular Biology, Iuliu Hatieganu University of Medicine and Pharmacy, 6 Louis Pasteur Street, 400349 Cluj-Napoca, Romania; 4Department of Prevention in Dentistry, Iuliu Hatieganu University of Medicine and Pharmacy, 31 Avram Iancu Street, 400083 Cluj-Napoca, Romania

**Keywords:** *Enterococus faecalis*, smear layer removal, ultrasonic activation, laser activation, sodium hypochlorite

## Abstract

This study aimed to evaluate the antibacterial effect of three final irrigation protocols and to compare their ability to remove the smear layer and debris from the root canal. Methods: Sixty-three single-rooted human teeth were inoculated with *Enterococcus faecalis* for 14 days. The teeth were divided into a positive control group (N = 3) and three treatment groups (N = 20) as follows: final irrigation with saline solution (control group), irrigation with 5.25% NaOCl ultrasonically activated with EndoUltra (EU), standard needle irrigation with Qmix 2in1 (Qx) and irrigation with 5.25% NaOCl activated using SiroLaser Blue (SB). The bacterial load was evaluated by analyzing the colony-forming units (CFU/mL). Selected specimens were split longitudinally and examined using scanning electron microscopy in order to determine the presence of a smear layer and debris. Statistical analyses were performed using one-way ANOVA and the Kruskal–Wallis rank-sum test. Results: Activation of NaOCl with EndoUltra or SiroLaser Blue was superior at reducing intracanal bacteria relative to standard needle irrigation with Qmix 2in1 solution (*p <* 0.05). Even though SiroLaser Blue showed the best results for removing the smear layer and debris, no significant differences were detected between the groups (*p* > 0.05). Conclusions: Final irrigation with 5.25% NaOCl ultrasonically activated using EndoUltra or SiroLaser Blue demonstrated a higher efficacy in bacterial reduction than standard needle irrigation with Qmix 2in1.

## 1. Introduction

The main targets of endodontic treatment are to prevent apical periodontitis when it is absent and to treat it when it is present by performing a proper shaping, cleaning and disinfection of the root canal system [1]. Most pathological changes that affect the pulp and the periapical tissues have microbial etiology [2]. *Enterococcus faecalis* (*E. faecalis*) is a facultative anaerobic microorganism that is highly resistant to conventional chemo-mechanical preparation and calcium hydroxide [3]. These characteristics seem to be related to its capacity to deeply penetrate dentinal tubules and endure extreme living conditions. *E. faecalis* is usually found in cases of treatment failure, with it being the most common microorganism detected in already treated teeth [4]. As a result, this particular bacterium was used in numerous studies that compared the efficiency of different irrigation solutions [5]. Mechanical instrumentation is definitely one of the paramount factors in reducing the bacterial load of infected root canals, although it is not completely effective at removing all bacteria and debris [1]. It was reported that large areas of root canal walls remain untouched, regardless of the instrumentation technique used [6,7]. This problem is mostly due to the complexity of root canal anatomy and instrument features. Furthermore, instrumentation of root canal walls will produce a large amount of smear layer and dentinal debris that will be pushed into the dentinal tubules and anatomical irregularities of the root canal system, where remaining bacteria are hiding in order to survive and multiply. This organo-mineral content layer works as a barrier for the intracanal medication and irrigants, which allows for some particular species to escape their action [8]. 

Endodontic irrigation has an essential role in root canal debridement and is considered to be the only way of cleaning confined areas that are unreachable by mechanical instrumentation. Due to its antimicrobial properties and ability to dissolve organic matter, sodium hypochlorite (NaOCl) is the most popular substance used in clinical practice. It was demonstrated that its antimicrobial and solving properties depend upon its concentration, pH, action time and temperature, as well as the method of activating it. Unfortunately, after traditional irrigation with a syringe and needle, many irrigated root canals still have detectable bacteria and dentinal debris in areas unreached areas by the irrigant [9,10]. To meet the challenges of cleaning root canals, a new solution, namely, QMix 2in1 (Dentsply Tulsa Dental, Tulsa, OK, USA), was introduced on the market for smear layer removal with antimicrobial efficiency. QMix contains ethylenediaminetetraacetic acid (EDTA), chlorhexidine (CHX), a detergent and deionized water. It was designed as a final irrigant in order to replace the 17% EDTA final rinse solution [11]. QMix has already demonstrated its ability to remove a greater amount of smear layer compared with an EDTA final rinse solution [12,13]. Furthermore, QMix seems to have an effective action on *E. faecalis* after a final rinsing procedure of up to 60–90 s/root canal [14,15,16]. 

Over the past few years, sonic or ultrasonic activation of NaOCl, as well as laser therapy, has gained ground in endodontic treatments, where they have a significant impact on reducing bacterial populations and eliminating hard tissue debris from the root canal by allowing the irrigant to penetrate in lateral canals, dentinal tubules and uninstrumented areas [17,18,19]. Passive ultrasonic irrigation (PUI) relates to the activation of the irrigant through the production of cavitation effects and acoustic microstreaming. This procedure can be performed either by using an ultrasonically oscillating file that is introduced into the canal after its shaping and after intracanal delivery of the irrigant with a syringe or by using an ultrasonic handpiece with continuous delivery of the irrigation solution [20]. The EndoUltra system (Vista, Racine, WI, USA) consists of a cordless ultrasonic handpiece with a titanium tip (20/02) that is capable of inducing acoustic micro-currents and hydrodynamic cavitation into the root canal through the propagation of ultrasonic waves with a 40 kHz frequency. This device claimed to improve the cleaning and disinfection of a root canal system, facilitating the upward debris removal, eliminating the biofilm and the smear layer from the canal walls at a higher rate, and making microorganisms more sensitive to NaOCl due to its temporary action on the bacterial cell wall and cytoplasmic membrane. However, there are few reported studies on the efficiency of this ultrasonic device in root canal disinfection [21,22]. 

Laser systems have been used in the dental field for years, with many of them being useful in endodontics by enhancing the agitation of the irrigant, and thus, intracanal disinfection [23]. The chromatic radiation emitted by lasers is absorbed by the irrigation solution causing vaporization and the development of vapor bubbles at the fiber tip. These bubbles expand and then collapse after the laser pulses have stopped. Subsequently, a laser pulse will generate smaller secondary cavitation bubbles and shock waves because of photomechanical and photoacoustic effects, resulting in photon-induced photoacoustic streaming. This phenomenon will determine the movement of the irrigant in the non-instrumented areas, which, due to its emitting light, causes chemical effects by damaging cell membranes, proteins, membrane lipids and nucleic acids of microorganisms [24,25]. One of the newest laser systems used in dentistry is the SiroLaser Blue (Dentsply Sirona, Bensheim, Germany). The system provides three different forms of lasers working at three different wavelengths: blue (445 nm), infrared (970 nm) and red (660 nm). The infrared diode was designed by the manufacturer to reduce bacterial levels from the root canal, even up to 1000 μm in the dentinal tubules. Blue laser light was already used in the reduction of periodontal pathogenic microorganisms, with no improvement in the antibacterial effect for 445 nm laser irradiation [25]. So far, there are no in vitro or in vivo studies on the antibacterial endodontic effect of SiroLaser Blue infrared wavelength. Moreover, information is still needed about the capacity of current devices to remove the endodontic biofilm during the final rinsing procedure.

Therefore, the purpose of the present study was to evaluate and compare the bacterial reduction capacity of three different final clinical irrigation protocols (ultrasonically activated NaOCl, standard needle irrigation with Qmix 2in1 solution and NaOCl activated by a laser) that are frequently used in endodontic therapy, as well as to evaluate the cleanliness of root canal walls after applying these disinfection techniques. The null hypotheses tested were that there were no differences between antibacterial efficacies and the cleaning ability of these final irrigation techniques

## 2. Materials and Methods

### 2.1. Specimen Preparation

This study was conducted on 63 monoradicular teeth (canines, lateral incisors, central incisors) that were free of caries, cracks or fractures and presenting no apical resorption, previous endodontic treatment or prosthetic reconstruction. All teeth were extracted for periodontal reasons, cleaned of tissue debris with ultrasonic scalers (Newtron Booster, Satelec-Acteon, Merignac, France) and then stored in a saline solution until their preparation. They were obtained for this study after receiving written consent under an ethically approved protocol (nr. 169) from the Ethical Board of the local University. Prior to the experiment, X-ray images of all teeth were taken from mesio-distal and bucco-oral orientations in order to confirm the presence of a single root canal and the absence of possible internal resorption. 

The access cavities were made using round diamond burs and an Endo Acces Bur (Dentsply Sirona, Ballaigues, Switzerland). The permeability of root canals was performed using a 10 MMC file (Micro Méga, Bensançon, France) until it was visible at the apical foramen level. Then, 1 mm from this length was removed to establish the working length. Ethylenediamine-tetraacetic acid gel 15% (CK Endo-Prep Gel EDTA 15%, Cerkamed, Stalowa Wola, Poland) was used during the pathfinding process. Root canal instrumentation was performed as follows: preflaring and glide path with OneFlare and One G files, respectively (Micro Méga, Bensançon, France), while shaping with One Curve file, #25/0.06 (Micro Méga, Bensançon, France) was undertaken as indicated by the manufacturer. Root canals were irrigated after the use of each endodontic file with 1 mL of 5.25% sodium hypochlorite (NaOCl) (Cloraxid, Cerkamed, Stalowa Wola, Poland) using a 5 mL syringe and a 30-gauge needle (Cerkamed, Stalowa Wola, Poland). Final irrigation after the instrumentation was done with 2 mL of 5.25% NaOCl. 

After the chemo-mechanical treatment, all teeth were placed in a phosphate buffer solution (PBS) and sterilized at 121 °C for 20 min (Lisa 300 sterilizer, W&H, Bürmoos, Austria) to ensure the sterility of the root canals before the bacterial inoculation. Sterility was tested using the following procedures: three teeth were randomly selected and each of them was transferred into a 15 mL sterile tube. In each of the three tubes, 10 mL of brain heart infusion (BHI, BioRad, Marnes la Coquette, France) was added and then the tubes were incubated at 37 °C for 7 days. After the first 48 h of incubation, 100 µL of BHI was inoculated on blood agar medium (Columbia agar with 5% sheep blood, BioRad, Marnes la Coquette, France) and the plates were incubated at 37 °C for 48 h under aerobic and anaerobic conditions. These procedures (100 µL of BHI from each tube inoculated on blood agar, followed by incubation of the plates at 37 °C for 48 h) were repeated over the next 3–7 days. No bacterial colonies grew on any of the plates. 

### 2.2. Root Canal Contamination

A suspension with an optical density of 1 McFarland (3 × 10^8^ colony-forming units (CFU)/mL) was prepared using a densitometer (DEN-1 McFarland Densitometer, Biosan SIA, Riga, Latvia) from 24 h colonies of *Enterococcus faecalis* ATCC-29212 (American Type Culture Collection, Manassas, VA, USA) grown on blood agar medium at 37 °C under aerobic conditions. Following sterilization, each tooth was inoculated with the bacterial suspension until the root canal was filled. The inoculation process was performed using a sterile insulin syringe with a 29 gauge needle (BD Micro Fine™ Plus, Becton Dickinson, Le Pont de Claix, France) and each tooth was placed in a single well of a non-treated, sterile, flat-bottom 12-well plate with a lid (Cell culture plate, 12-well, Eppendorf AG, Hamburg, Germany) in BHI for 14 days at 37 °C. Every 3 days, the BHI was replaced, thus providing an adequate development environment.

On the 14th day, the teeth were removed from the wells and their apex was sealed with a resin composite (Herculite, KerrHave SA, Bioggio, Switzerland) on a clean, sterile plate with the purpose of imitating the apical periodontium. In order to confirm the infection of all teeth with *E. faecalis*, after apical sealing, every root canal was filled with BPS. Then a sterile Hedström file 25.02 was used with back and forth movements (10 strokes) on the canal walls. A 1 mL insulin syringe was used to collect the canal content until 100 µL of the inoculum was obtained for each root canal [26]. The collected liquid from each specimen was inoculated on a blood agar medium and subsequently incubated at 37 °C under aerobic conditions. After 24 h, the number of CFU per mL was counted.

### 2.3. Disinfection Procedures

The final irrigation of the specimens was performed on the same day. Root canals were disinfected by using 3 different clinical procedures (20 teeth/group), for the final irrigation. Three teeth were kept for the positive control group, where a sterile saline solution was used to rinse the root canals (no disinfection procedure for this group). An experienced endodontist performed the procedures for groups 1 and 2, while for the third group, a clinician with experience in using SiroLaser Blue operated the device. 

**Group 1 (EU):** Root canals were passively filled with NaOCl 5.25% using a 30-gauge side-vented irrigation needle. Then, 3 mL of NaOCl was constantly delivered over 1 min in the coronal third of the root canal while the irrigation solution was ultrasonically activated with an EndoUltra tip. The latter was previously inserted down to 2 mm from the working length and then moved up and down into the root canal. A stopper was placed on the tip of the instrument at the established working length in order to ensure length control.

**Group 2 (Qx):** 1 mL of Qmix 2in1 (Dentsply, Tulsa, OK, USA) was constantly delivered in the root canal over 1 min using a 30-gauge side-vented irrigation needle. The needle was placed at 2 mm of the working length and moved in an apical-coronal direction in order to avoid its entrapment in the canal. For the length control, a stopper was placed on the irrigation needle.

**Group 3 (SB):** The irradiation of root canals was achieved with a pulsed diode laser (SiroLaser Blue, Sirona, Germany) at an infrared wavelength of 970 nm and maximum output power of 14 W. A new, sterile tip (EasyTip Endo 200 µm) was used for each tooth. A laser disinfection procedure was used in the next steps, as follows: 4 cycles of 10 s each of NaOCl 5.25% irrigation solution activated by laser, with a 5 s break between each cycle. First, the root canal was passively filled with NaOCl 5.25% using a similar irrigation needle as in the other 2 groups. Then, 3 mL of NaOCl 5.25% were constantly delivered for 1 min in the pulp chamber of each tooth. The irradiation was performed after each irrigation cycle, starting at the apex of the tooth and retracting 1/4 of the length of the root canal every second. 

At the end of the three final irrigation procedures, all specimens were irrigated again with sterile saline solution to remove the NaOCl and Qmix 2in1 from root canals. PBS was then passively introduced in each root canal (positive controls too) and pushed on the root canal walls (10 strokes) using a sterile Hedström file 25.02. The collection of the contained liquid was performed using the same protocol that was applied before the disinfection, followed by the inoculation of 100 µL of inoculum/specimen on a blood agar medium. The inoculated media were then incubated at 37 °C for 24 h. After the incubation, the post-treatment CFU were counted.

### 2.4. SEM Evaluation

In order to be evaluated using scanning electron microscopy (SEM), 6 random specimens were selected from each group (the saline positive control group was excluded). The teeth were sectioned longitudinally and a ditch was made along the long axis of the tooth without penetrating the root canal. The use of a chisel allowed for the separation of each tooth into 2 halves, which were examined using SEM for smear layer and debris evaluation in the middle and apical thirds. Given the high risk of contamination at this stage and in order to preserve the sterility of the root canals, these procedures were performed in a clean area and only sterile instruments were used, including disposable instruments. The slicing was made using a diamond cutting disc 345 22MM 0.25 (Yeti dental, Engen, Germany) fitted to a micro-motor. The transportation of the sliced teeth to the SEM laboratory was performed in sterile medical collection containers.

Each specimen was placed on an aluminum cylinder using carbon-made double-sided tape (Electron Microscopy Sciences, Hatfield, PA, USA) with a coating of colloidal silver for the enhancement of the conductivity. Afterward, the specimens were coated with a thin gold layer in a Polaron E-5100 plasma-magnetron sputtercoater (PolaronEquipment Ltd., Watford, Hertfordshire, UK) to enhance the electron signal reflected from the specimen surface in the electronic microscope. The specimens were examined using the scanning electron microscope Jeol JSM-25 (Jeol Ltd., Tokyo, Japan), with an electron acceleration voltage of 15–30 kV and different magnifications (45–2000×). The images were captured in a digital format with a Deben Pixie-3000 processor (Deben UK Ltd., Suffolk, UK). Three pictures were taken digitally from each part of a specimen, with a total of 9 pictures per specimen.

### 2.5. Image Analysis

Evaluations were performed to determine the remaining debris and smear layer from the selected specimens, with a 5-level scoring system for each [1], using a set of SEM pictures. Debris was defined as dentine chips and pulp particles loosely attached to the root canal wall. The scoring of the debris was performed using 450× magnification. A smear layer was defined, as proposed by the American Association of Endodontists’ (2003) Glossary Contemporary Terminology for Endodontics (2003), as a surface film of debris retained on dentine or other surfaces after preparation that consists of dentine particles, remnants of vital or necrotic pulp tissue, bacterial components and retained irrigant. The smear layer was scored at a magnification of 1000×. Hulsmann’s scoring system [1] taken into consideration for debris was as follows: score 1—clean root canal wall (just a few debris particles), score 2—a few small-sized accumulations of debris, score 3—more accumulations of debris that covered less than 50% of the analyzed surface, score 4—more accumulations of debris that covered more than 50% of the analyzed surface and score 5—complete or almost complete coverage of the analyzed surface.

For the smear layer, the scoring system [1] criteria were as follows: score 1—absence of smear layer (visibly open dentinal tubules), score 2—small quantity of smear layer (some open dentinal tubules), score 3—homogenous smear layer coverage of the root canal wall (only a few dentinal tubules open), score 4—homogenous smear layer coverage of the entire root canal wall (none of the dentinal tubules visibly open) and score 5—inhomogeneous smear layer completely covering root canal wall (none of the dentinal tubules visibly open).

The screenshots taken as part of the SEM analysis were taken of the same specimen region (medium and apical thirds), increasing the magnification on the different thirds of the root canal (Figure 1). Each of the SEM pictures was analyzed by 2 observers (L.C. and I.C.), who were different from those who irrigated the root canals or coded the teeth. When there was a disagreement between their evaluations, the lower score was chosen. The observers consisted of dentists who performed mostly endodontic procedures in their daily clinical activity. Both observers were trained in the score criteria before starting this procedure. 

### 2.6. Statistical Analysis

Descriptive statistics were computed and box plots were used in order to describe and visually compare the distribution of the investigated variables across groups (the saline positive control group was not used for the statistical analysis). Means and 95% confidence intervals for the means and standard deviations (SDs) of these variables were computed using the bootstrapping method (1000 samples/replications). The normality of distributions for dependent variables (*E. faecalis* colonies, smear layer and debris) was visually assessed by using Q–Q plots.

One-way ANOVA for independent samples was applied to compare the efficacy of bacterial reduction (metric variable) among the 3 final irrigation protocols, as well as the cleaning ability of those in the different analyzed root canal regions (medium/apical). The homogeneity of the variance was assessed using Levene’s test. Tukey’s post hoc multiple comparison test using Bonferroni corrections was performed for pairwise comparisons between the irrigation groups (EU-Qx; EU-SB; Qx-SB). Two-way ANOVA on ranks for the independent samples (Kruskal–Wallis) test was used to compare the effect of the 3 irrigation protocols on the debris and smear layer removal (rank variables) in the medium and apical third of the root canal after the SEM examination. Dunnett’s post hoc test was chosen for post hoc multiple comparisons between the amount of smear layer or debris that remained on the root canal walls after applying each final irrigation protocol. The level of statistical significance was set at α = 0.05. Graphical representations and analyses were performed using the statistical program JASP (JASP Team 2021, JASP v0.16).

## 3. Results

Means, standard deviations (SDs), and minimum and maximum values for the number of *E. faecalis* colonies that remained after applying each different final irrigation protocol are presented in Table 1. The positive control group presented bacterial growth before and after irrigation with saline solution. The highest CFU mean value was observed for Qx, where the final irrigation and the disinfection of root canals were performed with Qmix 2in1 solution. The lowest mean value was obtained for SB, where SiroLaser Blue was used to activate the NaOCl solution. 

The average percentage of bacterial reduction in CFU/mL within the root canal after final irrigation with the three different protocols is shown in Table 2. Activation of NaOCl with Endo Ultra and SiroLaser Blue had an equal effect on the reduction of *E. faecalis* populations with no significant statistical differences between them (*p* = 0.901). Both of these final irrigation techniques were statistically more effective (*p <* 0.05) than standard needle irrigation with Qmix 2in1 solution.

The SEM examination showed that all methods positively influenced the cleaning of the root canal walls, but none of them was able to completely remove the debris and smear layer. In the middle third of the root canal, the lowest mean values were obtained for Qx and SB (Figure 2). However, no statistically significant differences were detected when the three groups were compared two-by-two (*p* > 0.05—Kruskal–Wallis rank-sum test). In the apical area of the root canal, the best results for removing the smear layer and debris were found for SB. However, no statistically significant differences (*p* > 0.05—Kruskal–Wallis rank-sum test) were observed between the three groups. When the efficacy of the smear layer and debris removal in the medium and apical third of the root canal was compared for all three final irrigation protocols, the cleaning of root canal walls was better in the medium third than in the apical one (Table 3). However, no statistically significant differences (*p* > 0.05) were found between the two of them.

## 4. Discussion

Final irrigation is the paramount factor for the success of endodontic treatment since mechanical instrumentation and irrigation during this procedure were shown to be incapable of completely removing bacteria along the entire root canal and properly cleaning its walls [1]. Cleaning of the root canal system is directly influenced by the appropriate removal of debris and the smear layer since microorganisms may survive and multiply faster in these shelters, which are considered protection shields and substrates of nutrients that will allow bacterial colonization of dentinal tubules [27,28]. Moreover, *E. faecalis* is capable of similarly invading dentinal tubules, even when the smear layer remains on the root canal walls [29,30]. Passive ultrasonic irrigation (PUI) and laser activation as a final procedure before root canal filling were shown to be effective at removing debris and the smear layer [31,32]. Both techniques produce a circular motion of the fluid around the vibrating file or fiber, but at the same time, create millions of bubbles that collapse into the irrigation fluid. This physical phenomenon promotes a powerful movement of the irrigation solution into the root canal that will act on its walls, leading to smear layer and debris removal. 

The null hypotheses tested in the present study were partially accepted since the two new final irrigations techniques used (NaOCl activated using EndoUltra or SiroLaser Blue) had an equally significant effect on bacterial reduction, but when compared with conventional needle irrigation using Qmix 2in1 solution they showed statistically significant differences. However, the cleaning ability of all three final irrigation protocols tested in the present study was similar, with no significant statistical differences (*p* > 0.05) between them. These results agreed with previous studies that have confirmed the efficiency of the EndoUltra device in the eradication of bacterial biofilms [22,33]. The performance of PUI depends upon the acoustic streaming and cavitation taking place. These phenomena are influenced by the transmission of acoustic energy from the ultrasonically oscillating file into the irrigant, which is a fact that depends on the velocity and displacement amplitude of the tip’s file. For the cavitation effect to occur in the root canal, the file must vibrate at a displacement amplitude of at least 135 µm [34]. Considering what Ahmad et al. suggested [34], EndoUltra, with its total tip distance displacement of 319 µm at a frequency of 45 kHz and 154 µm at a frequency of 91 kHz, exceeds the distance threshold to achieve cavitation. Furthermore, the tip speed calculated for the EndoUltra was 14.5 m/s and 28.1 m/s, respectively, at its fundamental frequency, which are both greater than the 14.1 m/s cavitation threshold [35]. The performance of this device in producing cavitation can explain its efficiency, as demonstrated in the present study regarding the removal of *E. faecalis*. 

On the other side, the laser efficiency in endodontic final irrigation can be influenced by different parameters, including the time of application within the root canal, tip size and power [36]. The final disinfection protocol of one minute applied in this study and already recommended by the results of other similar studies in the literature [37,38] demonstrated a high rate of *E. faecalis* reduction (99.9%). At a lower wavelength (red light of 660 nm) but for an increased irradiation time (320 s), a diode laser with a power of 50 mW and a tip with 600 μm in diameter had the same efficiency as the SiroLaser Blue in eliminating this bacterium from the root canal [23]. Therefore, the use of tips with a greater size (600 μm vs. 200 μm) or longer exposure to irradiation (320 s vs. 60 s) in order to completely eradicate *E. faecalis* becomes questionable. An explanation of the very good results obtained when using SiroLaser Blue in the elimination of intracanal *E. faecalis* could also rely on the higher power and different wavelength used for its endodontic function in comparison to other laser devices [23,36,39]. Moreover, in the present study, during irradiation, the optical fiber was placed within the root canal at the apex of the tooth (at its total length), which allowed for the propulsion of the irrigant throughout the entire root canal system and the emission of light at the top of the tip but also on its sides, thus obtaining a wide distribution of infrared light and improving its antimicrobial effect. This could explain the slightly higher efficiency of this device in reducing root canal infection compared with ultrasonically activated irrigation, even though no statistically significant differences were found between them. Similar results for bacterial reduction were reported by Ordinalo-Zapata [40]. By contrast, Bago Juric et al. [36] did not find any difference in the reduction rate between PUI and laser-activated irrigation, even though the latter obtained the largest number of sterile samples after its use. The energy of the laser irradiation probably created a more powerful movement caused by the formation of secondary cavitation bubbles [36]. Then again, Xhevdet [33] observed that PUI had higher efficiency compared with laser waves, whose effectiveness reached that of standard irrigation with 2.5% NaOCl only after 5 min of use. 

The success of sodium hypochlorite solution in the reduction and elimination of *E. faecalis* was presented by a large number of studies [41,42,43] and depends on the concentration of undissociated hypochlorous acid in solution, retention time in the canal, time of direct contact with the cell wall and volume used [10]. It seems that the vital functions of bacteria are affected by hypochloric acid, leading to cell death. The concentration of 5.25% NaOCl used in our study was extremely efficient at destroying bacterial cells in less than 30 s [43]. Moreover, agitation of the irrigant using ultrasonic and laser activation increased the speed of tissue dissolution by NaOCl, particularly in the difficult-to-reach areas of the root canal [13] and significantly reduced bacterial levels, as already demonstrated in the literature [44].

Because of its strong antibacterial effect, good biocompatibility with minimal cytotoxicity [45] and capacity to remove more of the smear layer when compared with NaOCl irrigation alternated with EDTA solution [14,15,16], Qmix 2in1 was chosen as a one-step irrigant for the final conventional disinfection protocol. To our knowledge, this is the first study that compared the efficacy of EndoUltra and SiroLaser Blue activation techniques with Qmix 2in1 rinse solution against the intracanal *E. faecalis* population. The results showed significantly decreased mean values of CFU/mL on blood agar plates after each final irrigation protocol compared with before the treatment. The activation of 5.25% NaOCl final irrigation solution with SiroLaser Blue and EndoUltra demonstrated its superiority relative to a classical final rinse, although Qmix 2in1 alone was chosen as the irrigation solution for this latter one. Both activated irrigation techniques provided complete eradication of *E. faecalis* from the root canals of 15 samples for PUI and 17 samples for the diode laser. This improvement in bacterial reduction (or eradication in some cases) from the infected endodontic canals was in accordance with the results of other studies, where these kinds of devices were tested [7,36,42,46]. In the present study, all three final irrigation protocols lasted 1 min each. The antibacterial effect of QMix 2in1 increased with the time of its delivery into the root canal. Between 3 and 10 min, the killing ability of QMix 2in1 increased to 100% [47]. It was shown that at 3 min, 6% NaOCl killed more bacteria than Qmix 2in1. This meant that in 1 min, Qmix 2in1 would have a lower antibacterial effect than 5.25% NaOCl activated by SB or EU. Because Qmix 2in1 contains EDTA, CHX and a surface active agent that lowers the surface tension of this solution, it has a similar effect as EDTA in removing the smear layer and debris from the root canal wall. On the other hand, NaOCl by itself did not remove the smear layer and debris. This specific action on the debris and smear layer was due to its activation by SB/EU that produced cavitation bubbles and shock waves in the root canal via photomechanical and photoacoustic effects.

Culturing and counting CFUs was the first microbial evaluation method used in studies on root canal disinfection. Even though it is considered the gold standard method of determining the efficacy of root canal disinfection, the CFU method can be questioned regarding the objective presentation of the microbial load of the root canal as long as it depends on the ability to collect bacterial cells from the endodontic system, where numerous areas are not mechanically reachable [35,39]. In the present study, sampling was performed after filling the root canal with PBS and scraping the surface with a sterile Hedström file 25.02. This procedure allowed for the collection of both planktonic bacteria and microorganisms adhering to canal walls, but did not take into account the possible surviving bacteria in the isthmuses. The big difference in the CFU and SD was high for EU and Qx groups because of two outliers (measured after the final irrigation protocol; an outlier was considered a value above |2| standard deviations for each group separately): one belonged to the EU group and had a higher CFU value (43) and another one was in the Qx group (102). After testing for the equality of variance, Levene’s test was significant (<0.001). For this reason, we used bootstrapped post hoc comparisons (1000 replications), Dunnett’s test, and the Kruskal–Wallis test in our analysis, which all led to the same conclusions. To check for a possible impact of outliers, we performed the same analysis (a sensitivity analysis), but this time we excluded the two outliers. The new results did not change the initial conclusions.

Increased size and taper of the root canal preparation can influence the efficiency of endodontic irrigation, first by enhancing the removal of a larger amount of infected dentin, and second, by permitting the irrigation needle to penetrate deeper into the root canal [48]. When irrigation is performed with a syringe and needle, root canals that are prepared to a larger size (apical size 0.40) are cleaner than those prepared to a smaller size (apical size 0.20) [49]. The reason is that the irrigation needle cannot reach the apical third of the root canal and the disinfecting solution will clean beyond the tip of the needle only at a limited distance (1 to 3 mm), depending on the diameter of the needle tip [50]. In our investigation, the preparation of root canals was conservative (#25/0.06). This non-invasive instrumentation maintained the natural geometry of the root canal and did not affect the tooth fracture resistance [51]. Nevertheless, this conservative apical size was supposed to decrease the efficiency of traditional irrigation, as well as of PUI. The acoustic streaming of PUI may lose intensity because the tip of the instrument will not be able to oscillate freely, touching canal walls in the apical area. Yet, in the present study, there were no significant differences between the abilities of EndoUltra and SiroLaser Blue to reduce the *E. faecalis* population from the entire root canal, with their results being clearly superior compared with standard needle irrigation technique with Qmix 2in1 solution. An explanation could be that the EndoUltra file had a diameter at the top of 20/100 mm and a taper of 2%, which were both below the size and the taper of the prepared root canal. Furthermore, the EndoUltra thin titanium tip gave it great flexibility, allowing it to penetrate even into curved root canals, vibrating at a high frequency, and leading to increased streaming speed and stronger acoustic streaming. Similar results were already obtained by Lee et al., who reported that smaller preparations resulted in canals that were as clean as larger preparations when the irrigant was ultrasonically activated [48]. It was shown that laser-activated irrigation techniques do not need any particular enlargement of the apical third because the cavitation bubbles and shock waves produced by laser pulses will reach the end of the root canal with or without contact between its tip and canal walls [52]. This could explain the good scores obtained by SiroLaser Blue.

In the present study, the removal of debris and the smear layer was evaluated in the medium and apical thirds of the root canal using a numerical assessment of these areas, which were obtained after an SEM examination. Separate reference photographs were generated for the debris and smear layer after the SEM screening of each third of the canal wall and the area showing the greatest amounts of debris and smear layer was recorded and analyzed. There were no significant differences in the cleaning of the two areas of the root canal independent of the irrigation method used. These results are consistent with those of previous studies where no significant differences were observed between the removal of the smear layer and debris in the apical and medium thirds of the root canal after manual-dynamic irrigation and sonic activation [32,53]. However, it was also reported in the literature that smear layer removal using PUI was significantly superior to manual irrigation [32] and that laser-activated irrigation was superior to sonic and ultrasonic irrigation techniques [54]. Better cleaning of root canal walls was also observed for laser-activated irrigation in comparison with standard needle irrigation [14]. Despite that, in the present study, there were no statistically significant differences in the removal of the smear layer and debris between SiroLaser Blue, EndoUltra and manual irrigation with Qmix 2in1 solution, even though SiroLaser Blue obtained better scores. Concerning the Qmix 2in1 solution, the results of the present study were in agreement with those obtained by Eliot et al. [12], who applied the same evaluation criteria as the ones employed in the present research. They found a score of 3 in the medium third and of 4 in the apical third of the root canal. Apparently, the vertical continuous motion of the needle during manual irrigation promoted dynamic corono-apical circulation of the solution, ensuring good cleaning of canal walls. Previous studies demonstrated that Qmix 2in1 solution removes the debris and smear layer better than NaOCl due to its chemical composition [55,56]. This was the reason why, in the present study, Qmix 2in1 solution was used for traditional irrigation and NaOCl was activated using ultrasound and laser in order to see where the smear layer and debris will be better removed by activating them. The resulting superiority of SiroLaser Blue in cleaning root canals of the debris and smear layer found in this study should be confirmed using a larger sample size, and why not in narrower and curved root canals since more conservative endodontic rotary and reciprocating files are nowadays available in the dental field. 

## 5. Conclusions

Under the aforementioned experimental conditions, the results of the present study confirm that all three final irrigation protocols effectively removed the *E. faecalis* biofilm from the root canal. However, the efficacy of standard needle irrigation with Qmix 2in1 solution was surpassed by ultrasonically or laser-activated 5.25% NaOCl. In terms of the debris and smear layer removal, the activation of NaOCl irrigation solution with EndoUltra and SiroLaser Blue performed similarly to manual irrigation with Qmix 2in1 solution in both the medium and apical thirds of the root canal. The present results suggested that the SiroLaser Blue and EndoUltra systems are promising in canal disinfection, as well as in debris and smear layer removal.

## Figures and Tables

**Figure 1 materials-15-06688-f001:**
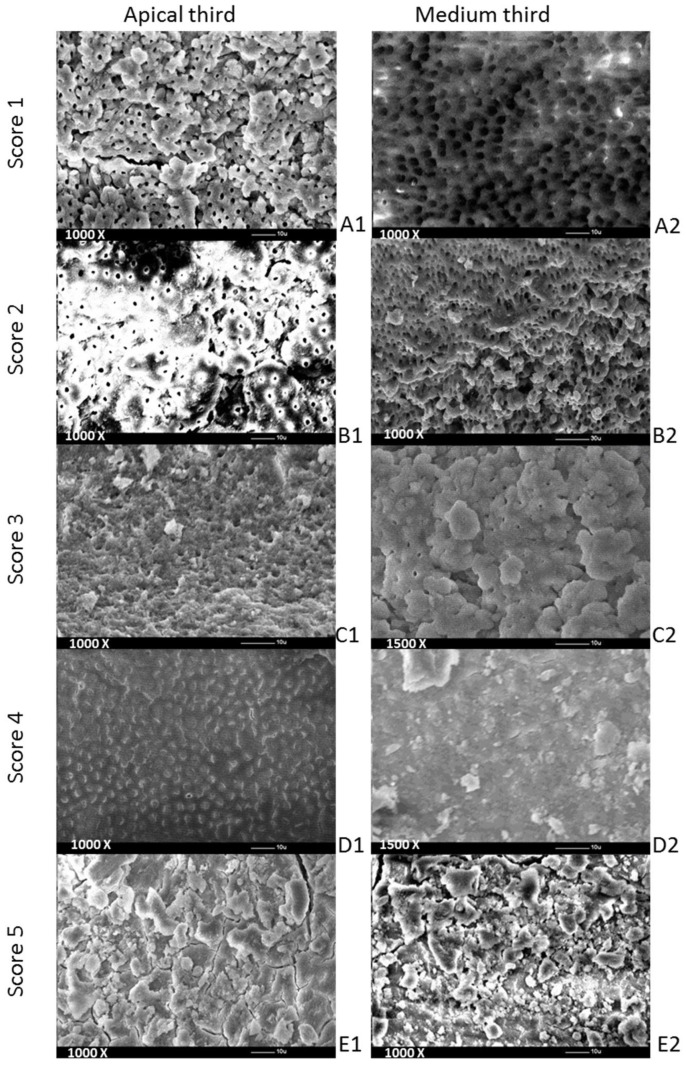
Representative SEM images (magnification of 1000/1500×) of smear layer scores from the medium and apical thirds of the root canal after exposure to the 3 final irrigation protocols tested: (**A1**) NaOCl 5.25% activated using SB, (**A2**) irrigation with Qx, (**B1**) NaOCl 5.25% activated using EU, (**B2**) NaOCl 5.25% activated using EU, (**C1**) irrigation with Qx, (**C2**) NaOCl 5.25% activated using SB, (**D1**) NaOCl 5.25% activated using EU, (**D2**) NaOCl 5.25% activated using EU, (**E1**) irrigation with Qx and (**E2**) NaOCl 5.25% activated using EU.

**Figure 2 materials-15-06688-f002:**
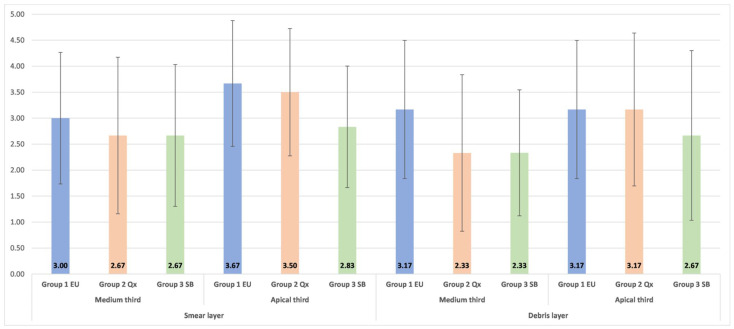
Means and standard deviations of debris and smear layer scores in the medium and apical third of the root canal.

**Table 1 materials-15-06688-t001:** Mean counts of *Enteroccus faecalis* CFUs 24 h post-treatment.

	*Mean*	*Standard Deviation (SD)*	*Min*	*Max*
Group 1 EU	2.85	9.76	0	43
Group 2 Qx	20	29.79	0	102
Group 3 SB	0.35	1.13	0	5

**Table 2 materials-15-06688-t002:** Reduction rate of *Enteroccus faecalis* in CFU mL^−1^ after irrigation.

	Bacterial Reduction (%)
Group 1 EU	98.1%
Group 2 Qx	86.45%
Group 3 SB	99.9%

**Table 3 materials-15-06688-t003:** Debris and smear layer scores in the medium and apical thirds of the root canal (mean and SD).

	N	Medium Third		Apical Third	
		Mean	Standard Deviation (SD)	Mean	Standard Deviation (SD)
Debris	18	2.833	1.339	3	1.414
Smear Layer	18	2.777	1.308	3.333	1.188

N—total number of SEM evaluated specimens.

## Data Availability

The data presented in this study are available on request from the corresponding author.

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
