# Peer review of "Effectiveness of Different Final Irrigation Procedures on Enterococcus faecalis Infected Root Canals: An In Vitro Evaluation"

_materials, 2022, doi:10.3390/ma15196688_

Round 1

Reviewer 1 Report

This manuscript investigates the antibacterial effect of and ability to remove smear layer and debris from root canal of 3 final irrigation ways. It’s a meaningful work from clinical perspectives. The manuscript has been well written. However, the experiments are insufficient to understand the insights and mechanism. And there are some problems in the manuscript. Some major comments are listed as following.

1. The authors only evaluated the CFU of Enteroccus faecalis after 24 h of incubation. What is the antibacterial efficiency of different irrigation methods with time?

2. Most of results in the manuscript were presented in Table. I strongly suggest the authors to show some of them as the graph. Because it is more clear for readers to understand.

3. What is the scale bar in Figure 1? It is hard to see.

4. The authors should discussed deeply the antibacterial mechanism of different irrigation methods.

5. Does debris and smear layer removal contribute to the antibacterial ability? For Group 3 SB, it showed better antibacterial efficiency compared to that of Group 2 Qx, however, there was a similar debris and smear layer scores between the two groups. The authors should discussed this point.

Author Response

Response to Reviewer 1 Comments

Point 1: The authors only evaluated the CFU of Enteroccus faecalis after 24 h of incubation. What is the antibacterial efficiency of different irrigation methods with time?

Response 1: The authors decided to evalaute the CFU after 24 h of incubation because Enteroccus faecalis is multipling easily and fast on blood agar (Patrick R. Murray, Ken S. RosenthaL, Michael A. Pfaller. Streptococcus and Enterococcus (Chapter 19). In Medical Microbiology, Eighth Edition, 2016, Elsevier, Philadelphia, PA; p.183-201.) In literature there are several protocols used for this kind of study, most of them choosing 24 or 48 h hours for the incubation process. (Xhevdet, A.; Stubljar, D.; Kriznar, I.; Jukic, T.; Skvarc, M.; Veranic, P.; Ihan, A. The disinfecting efficacy of root canals with laser photodynamic therapy. J. Lasers Med. Sci. 2014, 5, 19-26.

Pedullà, E.; Genovese, C.; Messina, R.; La Rosa, G.R.M.; Corsentino, G.; Rapisarda, S.; Ariaz-Moliz, M.T.; Tempera, G.; Grandini, S. Antimicrobial efficacy of cordless sonic or ultrasonic devices on Enterococcus faecalis-infected root canals. J. Investig. Clin. Dent. 2019,10, e12434)

Point 2: Most of results in the manuscript were presented in Table. I strongly suggest the authors to show some of them as the graph. Because it is more clear for readers to understand.

Response 2: We would like to thank the reviewer for the suggestion. We presented the results from both Tables 3 and 4 in one only graph (Fig. 2), in order to be more understandable for the readers.

Figure 2. Means and standard deviations of debris and smear layer scores in the medium and apical third of root canal.

Point 3: What is the scale bar in Figure 1? It is hard to see.

Response 3: The authors modified Figure 1 in order to be more understandable for the readers. The new version of this figure was inserted in the Manuscript.

Figure 1. Representative SEM images (magnification of X 1000) of smear layer scores from the medium and apical thirds of root canal.

Point 4: The authors should discuss deeply the antibacterial mechanism of different irrigation methods.

Response 4: Thank you for this remark. Authors included in the Discussion section a paragraph on the antibacterial mechanism of Qmix. For the antibacterial action of NaOCl an explanation was already given and it was also improved:

 “The success of sodium hypochlorite solution in the reduction and elimination of E. faecalis has been presented by a large number of studies [41-43] and depends on the concentration of undissociated hypochlorous acid in solution, retention time in the canal, time of direct contact with cell wall and volume used [10]. It seems that the vital functions of bacteria are affected by the hypochloric acid, leading to cell death. The concentration of 5.25% NaOCl used in our study is extremely efficient in destroying the bacterial cells in less than 30 seconds [43]. Besides, agitation of the irrigant by ultrasonic and laser activation increased the speed of tissue dissolution by NaOCl, particularly in the difficult to reach areas of root canal [13]  and significantly reduced  bacterial levels, as already demonstrated in literature [44].”

Point 5: Does debris and smear layer removal contribute to the antibacterial ability? For Group 3 SB, it showed better antibacterial efficiency compared to that of Group 2 Qx, however, there was a similar debris and smear layer scores between the two groups. The authors should discussed this point.

Response 5: Because Qmix contains EDTA, CHX and a surface active agent that lowers the surface tension of this solution, it has a similar effect as EDTA in removing smear layer and debris  from root canal wall. On the other hand, NaOCl by itslef does not remove smear layer and debris. This specific action on debris and smear layer is due to its activation by SB/EU that will produce cavitation bubbles and shock waves into the root canal by photomechanical and photoacoustic effects. In the present study all three final irrigation protocols lasted 1 minute each. It is also true that it was demonstrated that the antibacterial effect of QMix increases with the time of its delivery into the root canal. Between 3 and 10 minutes the killing ability of QMix increses with 100%. (Wang, Z.; Shen, Y.; Haapasalo, M. Effect of smear layer against Disinfection Protocols on Enterococcus faecalis-infected dentin. J. Endod. 2013, 39, 1395-1400.) It was proven that at 3 minutes 6% NaOCl killed more bacteria than Qmix. This means that in 1 minute Qmix would have a lower antibacterial effect than 5.25% NaOCl activated by SB or EU.

This explanation was inserted in the text, page 10.

Reviewer 2 Report

In the summary it is not clear the division in groups

In table 1, between groups, there is a huge difference in the numbers of CFU  and SD. Is the methology correct? Are the groups comparable?

Author Response

Response to Reviewer 2 Comments

Point 1. In the summary it is not clear the division in groups

Response 1: We thank this reviewer for his suggestion. We modified in the Abstract the division of the groups as below.

“Abstract: This study aimed to evaluate the antibacterial effect of 3 final irrigation protocols and to compare their ability to remove smear layer and debris from root canal. Methods: Sixty-three single-rooted human teeth were inoculated with Enterococcus faecalis for 14 days. The teeth were divided in a positive control group (N=3) and 3 treatment groups (N=20) as follows: final irrigation with saline solution (control-group), irrigation with 5.25% NaOCl ultrasonically activated with EndoUltra (EU), standard-needle irrigation with Qmix 2in1 (Qx) and irrigation with 5.25% NaOCl activated by SiroLaser Blue (SB). Bacterial load was evaluated by analyzing the colony-forming units (CFU/mL). Selected specimens were split longitudinally and examined by Scanning-Electron-Microscopy in order to determine the presence of smear layer and debris. Statistical analyses were performed using one-way ANOVA and Kruskal-Wallis rank-sum test. Results: Activation of NaOCl with EndoUltra or SiroLaser Blue were superior in reducing intracanal bacteria than standard-needle irrigation with Qmix 2in1 solution (p < 0.05). Even though SiroLaser Blue showed the best results in removing smear layer and debris no significant differences were detected among groups (p > 0.05). Conclusions: Final irrigation with 5.25% NaOCl ultrasonically activated by EndoUltra or SiroLaser Blue demonstrated a higher efficacy in bacterial reduction than standard-needle irrigation with Qmix 2in1. “

Point 2. In table 1, between groups, there is a huge difference in the numbers of CFU and SD. Is the methology correct? Are the groups comparable?

Response 2: This remark is true. There is a big difference in the numbers of CFU and SD is high for EU and Qx groups because of two outliers (measured after the final irrigation protocol; outlier = value above |2| standard deviations for each group separately): one belongs to EU group and had a higher CFU value (43) and another one was in Qx group (102). We tested for the equality of variance and Levene’s test was significant (<0.001). For this reason, we used in our analysis Bootstrapped Post-Hoc Comparisons (1000 replications), Dunnett test, and Kruskal-Wallis test which all lead to the same conclusions. To check for a possible impact of outliers, we performed the same analysis (a sensitivity analysis), but this time we excluded the two outliers. The new results did not change the initial conclusions.

Round 2

Reviewer 1 Report

The scale bars in Figure 1  are still unclear. The author should add them according to scale bars in SEM.

Author Response

Response 2 to Reviewer 1 Comments

Point 1: The scale bars in Figure 1  are still unclear. The author should add them according to scale bars in SEM.

Response 1: The authors modified the figure according to the suggestion of this reviewer. We hope that now the scale bars are clearer for the readers. Figure caption was also changed.

Fig. 1 Representative SEM images (magnification of X 1000/1500) of smear layer scores from the medium and apical thirds of root canal after exposure to the 3 final irrigation protocols tested: (A1) NaOCl 5.25% activated by SB, (A2) irrigation with Qx, (B1) NaOCl 5.25% activated by EU, (B2) NaOCl 5.25% activated by EU, (C1) irrigation with Qx, (C2) NaOCl 5.25% activated by SB, (D1) NaOCl 5.25% activated by EU, (D2) NaOCl 5.25% activated by EU, (E1) irrigation with Qx, (E2) NaOCl 5.25% activated by EU.

Reviewer 2 Report

Authors stated: "Response 2: This remark is true. There is a big difference in the numbers of CFU and SD is high for EU and Qx groups because of two outliers (measured after the final irrigation protocol; outlier = value above |2| standard deviations for each group separately): one belongs to EU group and had a higher CFU value (43) and another one was in Qx group (102). We tested for the equality of variance and Levene’s test was significant (<0.001). For this reason, we used in our analysis Bootstrapped Post-Hoc Comparisons (1000 replications), Dunnett test, and Kruskal-Wallis test which all lead to the same conclusions. To check for a possible impact of outliers, we performed the same analysis (a sensitivity analysis), but this time we excluded the two outliers. The new results did not change the initial conclusions."

I think that this point should be mentioned in the discussion section

Author Response

Response 2 to Reviewer 2 Comments

Point 1: Authors stated: "Response 2: This remark is true. There is a big difference in the numbers of CFU and SD is high for EU and Qx groups because of two outliers (measured after the final irrigation protocol; outlier = value above |2| standard deviations for each group separately): one belongs to EU group and had a higher CFU value (43) and another one was in Qx group (102). We tested for the equality of variance and Levene’s test was significant (<0.001). For this reason, we used in our analysis Bootstrapped Post-Hoc Comparisons (1000 replications), Dunnett test, and Kruskal-Wallis test which all lead to the same conclusions. To check for a possible impact of outliers, we performed the same analysis (a sensitivity analysis), but this time we excluded the two outliers. The new results did not change the initial conclusions."

I think that this point should be mentioned in the discussion section

Response 1: Thank you for your suggestion. We explained this point in the Discussion section pages 12 and 13.

“Culturing and counting CFUs was the first microbial evaluation method used in studies on root canal disinfection. Even though considered the gold standard method of determining the efficacy of root canal disinfection CFU method can be questioned in the objective presentation of the microbial load of root canal, as long as it depends on the ability to collect bacterial cells from the endodontic system, where numerous areas are not mechanically reachable [35,39]. In the present study sampling was performed after filling the root canal with PBS and scraping of the surface with a sterile Hedström file 25.02. This procedure allowed the collection of both planktonic bacteria and microorganisms adhering to canal walls, but didn’t take into account the possible surviving bacteria in the isthmuses. The big difference in the numbers of CFU and SD was high for EU and Qx groups because of two outliers (measured after the final irrigation protocol; outlier = value above |2| standard deviations for each group separately): one belongs to EU group and had a higher CFU value (43) and another one was in Qx group (102). After testing for the equality of variance, Levene’s test was significant (<0.001). For this reason, we used in our analysis Bootstrapped Post-Hoc Comparisons (1000 replications), Dunnett test, and Kruskal-Wallis test which all lead to the same conclusions. To check for a possible impact of outliers, we performed the same analysis (a sensitivity analysis), but this time we excluded the two outliers. The new results did not change the initial conclusions.”
